# Frequency and Intensity of Electrical Stimulation of Human Sympathetic Ganglia Affect Heart Rate Variability and Pain Threshold

## Sung-Hyoun Cho

Department of Physical Therapy, Nambu University, Gwangju 62271, Korea; geriatricpt1@gmail.com;
Tel.: +82-62-970-0232; Fax: +82-62-970-0492

**Featured Application:** This study evaluated the effects of electrical stimulation on the autonomic nervous system by evaluating heart rate variability (HRV) and the pain threshold in response to different conditions of interferential current (IFC) applied to the sympathetic ganglia. Different conditions of the frequency and intensity of electrical stimulation resulted in distinct changes in HRV and pain control duration. Therefore, an appropriate use of the frequency and intensity of electrical stimulation can be very important for its application in clinical treatments.

**Abstract:** The study relates to the selection of effective clinical treatments based on the changes associated with each electrical stimulation condition. The aim was to investigate the effects of electrical stimulation on the autonomic nervous system by evaluating the heart rate variability (HRV) and pain threshold in response to different interferential current conditions applied to the sympathetic ganglia. Forty five participants were randomly assigned to receive high frequency-low intensity (HF-LI), low frequency-high intensity (LF-HI), or high frequency-high intensity (HF-HI) electrical stimulation. We then used bipolar adhesive pad electrodes to stimulate the thoracic vertebrae T1–T4 for 20 min, and changes were evaluated before, immediately after and 30 min after electrical stimulation. Results revealed significant HRV immediately after HF-LI and LF-HI electrical stimulations. This present study finding of a reduction in HRV immediately after HF-HI electrical stimulation confirms HRV measurement reliability based on electrical stimulation parameters. Results revealed a significant increase in the pain threshold with HF-HI electrical stimulation than for the other conditions; there was also a shorter pain duration. The present study also showed a significant effect of the HF-LI and LF-HI conditions on the pain threshold immediately after electrical stimulation, but the results after 30 min only revealed significant changes in the LF-HI group, indicating a maintenance of the pain control period immediately and 30 min after electrical stimulation. Different conditions of electrical stimulation resulted in distinct changes in HRV and pain control duration.

**Keywords:** electric stimulation; ganglia; heart rate; pain threshold; sympathetic nervous system.

## 1. Introduction

Electrical stimulation is used in clinical practice to treat a diverse range of medical conditions, including skeletal and neurological disorders. Application of electrical current to the body can activate the membrane of nerve cells, changing their metabolic activity, and increasing blood circulation [1]. Since the stimulation sensation and penetration depth may differ among electrical stimulation application methods, they must be properly controlled to prevent accommodation and to achieve an effective treatment result [2].

Electrical stimulation has a good safety profile and can be applied to control pain [3]. Pain is determined by the responses of the nociceptive system to different stimuli. However, the responses to stimuli are subjective, because they may vary based on each individual's level of perception in the cerebral cortex [4]. The current conductivity is not uniform across all tissues at the time of electrical stimulation, hence, the actual patterns of maximum stimulation are diverse and highly irregular [3]. Nevertheless, pain is predicted to be correlated with the response to electrical stimulation owing to the fact that it is accompanied by multiple autonomic nervous system (ANS) symptoms [5]. Assessment of this pain threshold enables a further understanding of pain and changes in its symptoms [6]. Accordingly, the conductivity and excitability of the nerves have been measured quantitatively using electrophysiological methods in the clinic, while current threshold and variability in pain control have been used as objective assessment methods [4, 7]. Therefore, research is required to obtain empirical evidence for numerous stimulation variables, because the parameters of electrical stimulation associated with the mechanism of pain control activation can be divided into two main categories: Stimulated region and electrical characteristics [2].

Interferential current (IFC) therapy, which involves the application of medium-frequency currents at 1–100 kHz, is a method of transmitting burst frequency currents within the body's physiological range. IFC generates effective changes in the motor nerves because it does not cause the stimulated nerve fibers to adapt, and the extremely short pulse period used sends most of the currents directly to the tissues [2]. Liang et al. [8] and Picelli et al. [9] reported the following frequency-dependent changes occurring in the peripheral blood vasculature during electrical stimulation: A 2-Hz electrical stimulation is sufficient to suppress inflammation, increase cerebral blood flow and mitigate acute pain; a 100-Hz electrical stimulation affects the balance of central and local factors at a systemic level, thus preventing hyperactive bowel activity and blood pressure elevation. Kajbafzadeh et al. [10] also reported that IFC can relieve pain, reduce swelling, promote tissue recovery and relax contracted muscles by inducing vasodilation and increasing blood flow. In particular, medium-frequency currents have been used in the functional recovery of tissues by actively stimulating them and controlling associated pain (e.g., osteoarthritis pain, fracture pain, ischemic pain and mechanical pain) [11]. IFC is thought to exert its analgesic effects by increasing blood flow via the direct stimulation of the muscle fibers rather than the peripheral nerves, thus promoting the therapeutic process [12]. However, despite numerous studies demonstrating the therapeutic effects of IFC, no consensus has yet been reached regarding its analgesic mechanisms. Thus, it is crucial to determine the factors relevant to diagnosis, treatment evaluation and the effects of electrical stimulation [13].

The effects of the sympathetic nervous system are mediated by norepinephrine and epinephrine (noradrenalin and adrenaline) through three $\alpha$1-adrenergic receptors, three $\alpha$2-adrenergic receptors and three β-adrenergic receptors (β1, β2, β3) [14]. G protein-coupled receptors (GPCRs), such as the β-adrenergic and angiotensin II receptors, play crucial roles in regulating cardiac function and morphology. Their importance in cardiac physiology and disease is reflected by the fact that, collectively, they are the direct targets of over a third of the currently approved cardiovascular drugs used in clinical practice [15]. Beta blocker (β-blockers) induce reverse remodeling in a failing heart, improve survival, reduce the risk of arrhythmias, improve coronary blood flow (oxygen supply) to the heart and protect the heart against the cardio-toxic overstimulation by the sympathetic nervous system [14].

The physical reactions mediated through the ANS are manifested in response to external stimuli because of the crosstalk between the central nervous system and the ANS. Additionally, the physiological changes necessary for sustaining life are unconsciously or reflexively controlled, which is reflected in the cerebral cortex [16]. Many research efforts have been made to quantify this process using electrocardiography and cardiovascular, sweat and vasomotor function tests, and to explain the underlying mechanisms for the pathological aspects of the central neural activity, in order to set up efficient treatment strategies. In particular, many studies have been conducted to quantify changes in the ANS via heart rate variability (HRV) analysis [17].

HRV analysis, an important tool for assessing heartbeat changes induced by physiological factors, provides clues for diagnosing ANS-related abnormalities and identifying various changes in the physical health status [18, 19].

The hypothesis of this study was that the HRV and pain thresholds between each group would differ according to the frequency and intensity of the electrical stimulation. The aim of this study was to explore the effects of electrical stimulation on the ANS by evaluating the HRV and pain threshold in response to the different conditions of IFC applied to the sympathetic ganglia. This article provides valuable basic data for the selection of effective clinical treatments based on the changes associated with each condition of electrical stimulation used in this study.

## 2. Materials and Methods

This study involved a parallel design to compare the means of three groups, and the analysis method used for the main evaluation variables was a two-way repeated measures Analysis of Variance (ANOVA). We included 45 healthy students living in Kwangju city, Republic of Korea, in the study. The participants were provided with sufficient explanation about the study, and all voluntarily agreed to participate in the experiment. The sample size was estimated from power calculations using a commercially-available software package (G-Power). Assuming a power of 90%, a type 1 error ($\alpha$) of 5%, a type 2 error ($\beta$) of 10%, and a confidence interval of 95%, we calculated a sample size of 15 per group [20, 21]. Exclusion criteria were as follows: Medical history and vital signs (blood pressure, pulse, body temperature, respiratory rate) that may be associated with changes in blood flow; conditions that may influence the blood test and experimental results; metal prosthetic implants; neurological or brain disorders; and unsuitability for electrical stimulation intervention or other. In order to control for factors that may influence the results, the participants were instructed to refrain from vigorous exercise that may influence their heart rate variability (HRV). They were also required to abstain from smoking and the drinking of alcohol and coffee, which may influence the autonomic nervous system (ANS), 1 h prior to the experiment. In addition, the participants were given instructions about other factors that may affect the results (e.g., talking, coughing and deep breathing), and were requested to avoid such behaviors during the measurements. Preliminary measurements were conducted, and subjects who had a large HRV that deviated from the standard value due to excessive sweating on the palms, or cold hands, were excluded. The participants were randomly assigned to receive high frequency-low intensity (HF-LI), low frequency-high intensity (LF-HI), or high frequency-high intensity (HF-HI) electrical stimulation (n = 15 per group). For the analysis, the three groups were considered to have similar general characteristics (Table 1).

A flowchart of the study procedure is shown in Figure 1. The study protocol was approved by the institutional review board of Nambu University in Kwangju, and the participants provided written consent after receiving verbal and written information about the study.

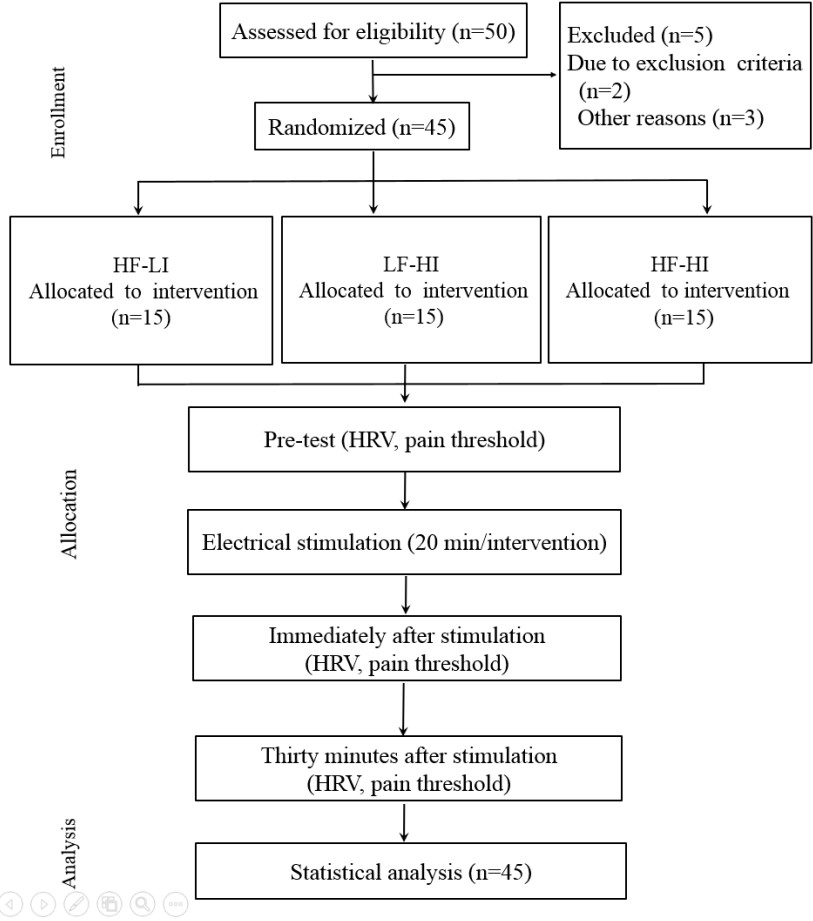

**Figure 1.** Flowchart of the study procedure.

HF-LI: high frequency-low intensity, LF-HI: low frequency-high intensity, HF-HI: high frequency-high intensity

**Table 1.** General characteristics of the study participants.

| Characteristics | | HF-LI | LF-HI | HF-HI | F | P |
|---|---|---|---|---|---|---|
| **Sex** | Men | 7 | 8 | 9 | - | |
| | Women | 8 | 7 | 6 | | |
| Age (years) | | 22.00 ± 1.41 | 22.45 ± 1.44 | 22.64 ± 1.63 | 0.527 | 0.596 |
| Height (cm) | | 166.27 ± 7.46 | 167.09 ± 8.50 | 170.64 ± 8.08 | 0.615 | 0.547 |
| Weight (kg) | | 62.55 ± 4.55 | 60.82 ± 4.08 | 65.09 ± 5.93 | 0.512 | 0.640 |

Note: Data are expressed as the number or mean ± standard deviation; HF-LI: high frequency-low intensity, LF-HI: low frequency-high intensity, HF-HI: high frequency-high intensity.

The Endomed 582 (Enraf Nonius, The Netherlands) was used as an interferential current (IFC) electrical stimulator. Each intervention lasted 20 min. Pad electrodes (4 × 4 cm, Protens Electrodes, Bio Protech Inc., Korea) were placed on the transverse process at a distance of 2 cm from the left side of the spinous process of the thoracic vertebrae T1–T4. To ensure the psychological stability of the subjects, these subjects were asked to lie down on a bed for 10 min before intervention [11]. The conditions used for the HF-LI, LF-HI and HF-HI groups were 100 bps/10–20 mA, 5 bps/45–50 mA, and 100 bps/80–90 mA, respectively. The amplitude refers to the stimulation's applied wave voltage or current intensity. In this study, electrical stimulation produced three types of excitatory responses: Sensation, motion, and pain.

The sensation stimulation had a current intensity range of 50–100 bps, which can be felt by the subject; the motion stimulation had a range of 2–10 bps, which produces a muscular contraction; and harmful stimulations were delivered at 1–5 bps or 100 bps or higher at a high frequency, and the stimulation period lasted at least 1 s [22].

HRV refers to the natural variation in the time between each heartbeat or the beat-to-beat alterations in heart rate (inter-beat interval) in 1 min. HRV analysis methods are classified into frequency domain analysis and time domain analysis methods. The frequency domain analysis of HRV displays the activities of the sympathetic and parasympathetic nervous systems in different frequency ranges [23]. In the time domain analysis method, the beat-to-beat interval variations, i.e., the R-R interval differences, are analyzed, whereby the standard deviation (SD) of the total R-R intervals reflects the information on the ANS's ability to regulate homeostasis and the stability of the cardiovascular system [24]. The R-R distance in HRV is determined under the influence of breathing, vasomotion, sympathetic and parasympathetic nerves, pressure receptors, chemoreceptors and the renin–angiotensin system, among others, which produce stimulations that have regular wavelengths [25]. Specifically, with the measurement and quantification of fluctuations of the R-R interval, a correlation with a particular periodicity has recently been discovered between HRV and autonomic nerve activity, indicating that HRV is a biological signal that can be measured non-invasively and used as an objective index for a predictive assessment of ANS function and activity [26]. By displaying the R-R interval variations as the mean and standard deviation of the results of a 3-min R-R interval measurement, these can serve as an objective indicator obtained from the statistical observation of the R-R interval variability.

The measuring device was the QEEG-8 System (Poly G-I, Laxtha Inc., Korea) and measurements were made when R waves were visible on the electrocardiogram at both poles. When measuring HRV, the laboratory temperature was maintained at 20–25 °C to prevent the effects of temperature on the autonomic nerves. The experiment was conducted in a stable space with blocking of external noise, and the measurements were taken while the subjects were lying down. Furthermore, measurements were determined between 180 and 300 s to minimize the effects of any artifacts that could influence the data in the case of a short or long measurement period. The measurements were taken at 220 s before, immediately after and 30 min after the electrical stimulation, and the analysis was conducted on the 20–180-s period, subtracting 20 s each at the beginning and at the end.

The Endomed 581 (Enraf Nonius) was used to quantify the changes in the pain threshold. It was applied to the left radialis muscle, which is visible when the elbow is bent. A point was selected such that the greatest response was achieved to pulsating rectangular waveform currents applied through the active electrode, and when placing the electrode against the most protruded part, a 200-ms pulse length and a 2,000-ms pulse interval were used. Probe electrodes were used as the stimulating electrodes; the intensity was increased by 0.2 mA in 1-s intervals, and the current intensity at the point when the subject reported a stinging pain sensation was taken as the pain threshold [27]. The measurement was repeated three times to calculate an average value.

All data were analyzed using SPSS version 23.0 for Windows (IBM, Armonk, NY, USA). The Shapiro–Wilks test was used to check for the normal distribution of the general characteristics data and values measured. As a result, a normal distribution was confirmed. Then, ANOVA was used to compare the general characteristics between groups. Two-way repeated measures ANOVA was used to investigate changes in the HRV and pain threshold for each group over time. The significance of values measured was analyzed by post-hoc analysis using Tukey's method. The significance level ($\alpha$) of the statistical analysis was set at 0.05.

## 3. Results

Changes in HRV are shown in Table 2. Assuming sphericity, based on the absence of a statistically significant difference in Mauchly's test of sphericity (Mauchly's W), the within-subject effects were tested, showing a time-dependent significant difference in HRV between immediately after and 30 min after electrical stimulation, with a significant reciprocal interaction between time and group.

Specifically, there was a significant difference between the HF-LI and HF-HI groups immediately after electrical stimulation, and between the HF-LI and LF-HI groups 30 min after electrical stimulation ($p < 0.05$).

Changes in pain threshold are shown in Table 2. Mauchly's test of sphericity was statistically significant. A multivariate analysis revealed a time-dependent significant difference in the pain threshold between before, and 30 min after, electrical stimulation. There was also a significant interaction between time and group. Specifically, there was a difference between the HF-LI and HF-HI groups, and between the LF-HI and HF-HI groups immediately after, and 30 min after, electrical stimulation ($p < 0.05$).

**Table 2.** Changes in heart rate variability (HRV) and pain threshold.

| Variable | Group (n = 15 Per Group) | Time From Electrical Stimulation | | | F (p) | | |
|---|---|---|---|---|---|---|---|
| | | Before | Immediately After | Thirty Minutes After | Group | Period | Group × Period |
| HRV (Hz) | HF-LI | 69.94 ± 6.88 | 72.54 ± 6.10 | 70.99 ± 6.28 | 4.10 (0.02*) | 8.57 (0.00*) | 2.94 (0.02*) |
| | LF-HI | 65.88 ± 2.90 | 68.42 ± 3.31 | 65.69 ± 3.22 | | | |
| | HF-HI | 66.46 ± 3.29 | 66.43 ± 3.19 | 66.72 ± 4.92 | | | |
| Pain threshold (mA) | HF-LI | 0.92 ± 0.24 | 1.07 ± 0.32 | 1.09 ± 0.24 | 6.81 (0.04*) | 37.82 (0.00*) | 4.99 (0.00*) |
| | LF-HI | 0.90 ± 0.27 | 1.23 ± 0.34 | 1.31 ± 0.24 | | | |
| | HF-HI | 1.09 ± 0.28 | 1.73 ± 0.59 | 1.68 ± 0.40 | | | |

Note: Data are expressed as mean ± standard deviation; HF-LI: high frequency-low intensity, LF-HI: low frequency-high intensity, HF-HI: high frequency-high intensity. *$p < 0.05$.

## 4. Discussion

The dynamic evaluation of the ANS based on HRV analysis is important in the treatment of complications caused by ANS abnormalities [18]. As a diagnostic parameter for measuring imbalance in the ANS, the HRV analysis method for measuring cardiovascular ANS activities by observing the R-R interval variations on the electrocardiogram (ECG) has been used for the clinical evaluation of the ANS conditions of patients, to determine the balance between the sympathetic and parasympathetic nervous systems and the activation level, as well as for evaluating diagnostic markers for a given disease [19]. HRV is a determinant factor for evaluating the scope of changes in the ANS based on the beat-to-beat interval changes. Heartbeat is determined by the ANS regulatory function of the sinoatrial node, for the maintenance of homeostasis of the body and for spontaneous activity [28]. Sensory nerves aroused by electrical stimulation are known to influence the cardiovascular system via the ANS [19]. The results of the present study support previous research that found significant HRV immediately after HF-LI and LF-HI electrical stimulations [23]. However, Wang et al. [29] showed decreased activity in the sympathetic nerves after electro-acupuncture stimulation, which was accompanied by a decrease in HRV and pain in healthy men. The reduction of HRV immediately after HF-HI electrical stimulation in the present study is similar to previous research results, confirming the reliability of HRV measurement results based on electrical stimulation variables [30].

It should be noted that the results of this study could not directly demonstrate whether the electrical stimulation induces HRV via changes in the ANS. Because spectral analysis of heart rate and blood pressure changes are indices induced by the sympathetic nervous system, Parati et al. [31] questioned the validity of the use of these indicators alone for the assessment of cardiovascular regulation. Although we were able to determine that HF-HI electrical stimulation decreased HRV, the significant increase in the pain threshold observed also raises the same questions as in previous research, because it could be explained by a simple change in the ANS.

Fisher et al. [17] proposed that detailed research should be performed to elucidate the pathological mechanism of central sympathetic nerve activity and to develop an effective treatment strategy.

Pain types are assessed based on the theory that pain uses a specific pathway in which a single nerve cell forms a pattern within a complex neural network, taking information from multiple nerve cells [32]. The stimulation currents are normalized so that the pain control effect due to electrical stimulation is not affected by changes in tissue resistance; therefore, reliable measurement values can be obtained by maintaining a current that is invariant to changes of resistance [33].

However, electrical stimulation can also render negative results, regardless of pain control effects, such as changes in blood flow or increased release of neurotransmitters, such as bradykinin and histamine, due to pain induction and increased sympathetic nerve activity [34, 35]. This is why the mechanism for pain control can be explained based on frequency-intensity current parameters [36]. Platon et al. [37] claimed that when HF-HI electrical stimulation collides with stimulation from peripheral areas to block pain transmission from peripheral areas, the conductivity speed of the peripheral nerves is reduced, and the temporary block of afferent conductance in A-δ and C fibers acts to cause pain. The results of the present study also confirmed this mechanism: Although there was a significant increase in the pain threshold upon HF-HI electrical stimulation compared to the other conditions, there was also a shorter pain duration. By contrast, the effects of low-frequency electrical stimulation on pain were not local, had a long duration and were sometimes accompanied by sleepiness or calmness [38].

When performing electrical stimulation according to frequency, Liang et al. [8] reported that a frequency of 2 Hz was effective in relieving acute pain, and that 1–10 bps had stimulating effects on the motor nerves and tissues, leading to an increase in strong vasodilatory substances. Moreover, the combination of LF and HI electrical stimulation, which shows an increase in deep pain and endorphin secretion, had relatively slower pain induction but longer pain duration; therefore, this type of stimulation is often used for chronic pain [39]. On the other hand, it has been reported that HF and LI transcutaneous electrical nerve stimulation (TENS) treatment activates afferent nerve tissues with a larger diameter, and the sensation threshold emerges rapidly, but the duration is short [5]. The present study also showed a significant effect of the HF-LI and LF-HI conditions on the pain threshold immediately after electrical stimulation, but the results after 30 min showed there were significant changes only in the LF-HI group, indicating maintenance of the pain control period immediately after, and 30 min after, electrical stimulation. Chang et al. [40] showed that high-frequency electrical stimulation causes the release of β-endorphin by inhibiting serotonin, which has a top-down control effect, and the blocking of harmful information suppresses the pain-related activity of nerve cells within the spinal cord, leading to a reduction in pain and a decrease in the length of pain persistence. We consider that the difference in the pain threshold duration observed in the present study is due to the same mechanism.

Bergadano et al. [41] reported that the intensity of the electrical stimulation acts as an important factor in the pain threshold increase. Chen & Johnson [42] reported that subjects receiving strong, painless, electrical stimulation had a greater increase in their pain threshold. Claydon's research also indicated a significant effect in segmented stimulation at high intensity [43]. This could occur because when stimulation signals are sent to the central nerves by the physiological transmission systems for electrical signals, endogenous opioid secretion is increased, and μ receptors in the spinal cord are activated, producing a pain effect, blocking the pain transmitted to the higher central nervous system, and activating descending pain functions to suppress pain [44]. Therefore, when performing electrical stimulation for pain control, an important objective should be to control abnormal increases in harmful sensations [42]. Thus, future studies should focus on the persistence of treatment effects by broadening the range and making finer distinctions in the measurement of experimental pain thresholds.

The method of raising the electrical stimulation intensity and of considering the intensity at which pain first occurs as the threshold, allows for easy measurement, and the results can be quantified immediately.

This method has the further advantage of enabling pain measurements both before and after the treatment, such that the pain treatment effects can be accurately assessed. However, since the subjects have to evaluate the extent of the pain themselves, it is a subjective measurement. Experimental pain in healthy subjects is explained by the suppression of harmful stimulation [45], and the experimentally-induced pain is temporary [46]. Nevertheless, pain models in clinical experiments are important tools for researching pain mechanisms. Accordingly, clinical pain models have considerable potentials, compared to environmental situations, and it is thought that they should be further normalized with respect to trauma and the individual subject [47].

The limitations of this study can be assessed as changes in the ANS in vasoconstriction and blood flow rate, due to decreased HRV in the group at the level of the harmful stimuli, but a significant increase in pain may require additional physiological and functional anatomy studies to be described simply by changes in the ANS. Further research will establish the link between pain control and the ANS to prepare indicators for the improvement of autonomic nerves by variables of various conditions and complex electrical stimulation variables for treatment. A study using different electronic stimulation variables based on specific situations and purposes should be conducted for application in patients during clinical trials; this study was conducted on healthy adults. However, the results of this study are also useful for clinical applications in future; this study comprised selected examinations, based on the various types of patients encountered, for use in future clinical trials.

## 5. Conclusions

Different conditions of electrical stimulation resulted in distinct changes in HRV and pain control duration. Appropriate use of the frequency and intensity of electrical stimulation can be very important for physical therapy outcomes. Future studies should continue to focus on the stimulation of the spinal cord segments that affect the internal organs in more detail, and to comprehensively compare them with various examinations.

**Author Contributions:** S-H. C.; project administration, performed the experiments and data analysis, writing original draft.

**Funding:** This work was supported by the National Research Foundation of Korea (NRF) grant funded by the Korea government (MSIT) (No. 2017R1C1B5076499).

**Acknowledgments:** I would like to thank Editage (www.editage.co.kr) for English language editing.

**Conflicts of Interest:** The author declares no conflict of interest

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
