# Peer review of "Frequency and Intensity of Electrical Stimulation of Human Sympathetic Ganglia Affect Heart Rate Variability and Pain Threshold"

_applsci, doi:10.3390/app9214490_

Round 1
Reviewer 1 Report
The authors have addressed all of my concerns adequately. No further comments.
Author Response
I attached response file.

Reviewer 2 Report
Study of 45 patients receiving HF-LI, LF-HI, or HF-HI electrical stimulation at T1-T4 sympathetic ganglia on HRV and pain threshold. Reasonable study. Should clearly state effects of frequency and intensity of HRV in the abstract by intervention (rather than "significant interaction between group A and group B"). Should provide premise and next steps for using results for practical application. Were results transient or persistent, and how long did they last?
Author Response
I attached response file.

This manuscript is a resubmission of an earlier submission. The following is a list of the peer review reports and author responses from that submission.
Round 1
Reviewer 1 Report
This study is well conducted and the manuscript reads well overall. I only have a couple of comments:
1) The limitations of the study should be discussed further in "Discussion".
2) A couple of important reviews on the role of the sympathetic nervous system in heart physiology and disease should be cited and mentioned in "Introduction": Eur J Pharmacol. 2015;763:143-8; Circ Res. 2013;113:739-53.